# Conditional deletion of WT1 in the septum transversum mesenchyme causes congenital diaphragmatic hernia in mice

Rita Carmona[1,2], Ana Cañete[1,2], Elena Cano[3], Laura Ariza[1,2], Anabel Rojas[4,5], Ramon Muñoz-Chápuli[1,2]*

[1]Department of Animal Biology, University of Málaga, Málaga, Spain; [2]Andalusian Center for Nanomedicine and Biotechnology (BIONAND), Málaga, Spain; [3]Max Delbrück Center for Molecular Medicine, Berlin, Germany; [4]Andalusian Center of Molecular Biology and Regenerative Medicine (CABIMER), Sevilla, Spain; [5]Centro de Investigación Biomédica en Red de Diabetes y Enfermedades Metabólicas Asociadas (CIBERDEM), Sevilla, Spain

**Abstract** Congenital diaphragmatic hernia (CDH) is a severe birth defect. Wt1-null mouse embryos develop CDH but the mechanisms regulated by WT1 are unknown. We have generated a murine model with conditional deletion of WT1 in the lateral plate mesoderm, using the G2 enhancer of the *Gata4* gene as a driver. 80% of G2-*Gata4*[Cre];*Wt1*[fl/fl] embryos developed typical Bochdalek-type CDH. We show that the posthepatic mesenchymal plate coelomic epithelium gives rise to a mesenchyme that populates the pleuroperitoneal folds isolating the pleural cavities before the migration of the somitic myoblasts. This process fails when Wt1 is deleted from this area. Mutant embryos show Raldh2 downregulation in the lateral mesoderm, but not in the intermediate mesoderm. The mutant phenotype was partially rescued by retinoic acid treatment of the pregnant females. Replacement of intermediate by lateral mesoderm recapitulates the evolutionary origin of the diaphragm in mammals. CDH might thus be viewed as an evolutionary atavism.

*For correspondence: chapuli@uma.es

**Competing interests:** The authors declare that no competing interests exist.

## Introduction

Congenital diaphragmatic hernia (CDH) is a severe birth defect, characterized by incomplete formation or muscularization of the diaphragm and, as a consequence, herniation of the stomach, intestines, liver or spleen into the pulmonary cavities, leading to pulmonary hypoplasia. CDH occurs approximately in 1 out of 3000 births, accounting for about 8% of all severe congenital anomalies. 80–90% of all the cases are posterolateral hernias, also known as Bochdalek-type CDH, characterized by a defect in the postero (dorsal in mice) lateral area of the diaphragm. In most cases (>85%), this defect is located at the left side (*Pober et al., 2010*). The classical hypothesis is that Bochdalek CDH is due to a failure of the fusion of pleuroperitoneal folds (PPFs) with the septum transversum (ST). PPFs are lateral rims of tissue connecting caudally with the nephric ridges and anteriorly with the ST (*Mayer et al., 2011*). Some authors describe that this fusion rather occurs with the posthepatic mesenchymal plate (PHMP), an accumulation of mesenchymal cells derived from the ST and located in the posterodorsal margin of the liver lobes (*Iritani et al., 1984*). Although all these tissues form an anatomical continuum, we will refer to the posterior, dorsolateral areas of the liver as PHMP, and PPFs to the tissue folds located in the dorsal part of the coelomic cavity, marking the limit between the peritoneal and the pleural cavities.

The clinical aspects of CDH are well studied but its etiology is still poorly known (*Klaassens et al., 2006*). Animal models are therefore very valuable in order to investigate the

cellular and molecular processes leading to CDH in humans. Recently, it has been described in mice that deletion of GATA4 in the PPFs mesenchyme using a *Prrx1*[Cre] driver leads to the development of amuscular, weak areas in the diaphragm and CDH (*Merrell et al., 2015*). This paper assumes that Prrx1 is expressed in the mesenchyme of the pleuroperitoneal folds (PPFs) but not in the ST, and in fact, it suggests that the ST contributes minimally to the definitive diaphragm. However, this report does not distinguish between PPFs and PHMP and the phenotype does not display diaphragmatic discontinuities, neither does it show a prevalence on the left side, which are common features in human CDH.

The retinoic acid (RA) signaling pathway has also been involved in diaphragmatic development. A classical animal model for CDH consists in the treatment of pregnant rats with nitrofen, a substance that inhibits the synthesis of RA. It has been shown that nitrofen treatment leads to reduced size of the PPFs/PHMP, especially in the left side, decreasing cell proliferation in this area (*Clugston and Green, 2007*; *Clugston et al., 2010*) and also decreasing expression of WT1 and GATA4 (*Dingeman et al., 2011*, *2013*). RA treatment in this model partially rescues the pulmonary hypoplasia defect (*Montedonico et al., 2008*; *Sugimoto et al., 2008*). The phenotype of this animal model is thus, quite similar to the human one.

To date, only a few examples of CDH in human have been associated to mutations in WT1 locus, such as the Denys-Drash, Meacham and WAGR syndromes (*Scott et al., 2005*; *Suri et al., 2007*; *Antonius et al., 2008*). WT1 is expressed in the coelomic epithelium of the septum transversum of mouse embryos by the stage E9.0. Loss of function of the Wilms' tumor suppressor gene *Wt1* in mice leads to defective diaphragms (*Kreidberg et al., 1993*; *Clugston et al., 2006*). However, the mechanism by which WT1 contributes to diaphragm development is still unknown.

To study the role of WT1 in CDH we performed loss-of-function experiments by conditionally inactivating the *Wt1* gene in the ST/PHMP/PPFs mesenchyme. The use of WT1 conditional knockout overcomes the early embryonic death caused by systemic deficiency of WT1. We used a driver based on the G2 enhancer of the *Gata4* gene (*Rojas et al., 2005*). This enhancer drives expression of *Gata4* in the lateral plate mesoderm from the stage E7.5, and by the stage E9.5 is active in the septum transversum and proepicardium, ceasing its activity by E12.5 The activity of this enhancer is completely absent in the intermediate mesoderm. Our findings indicate that WT1 is involved in the generation of the mesenchyme of the ST/PHMP/PPFs continuum through epithelial-mesenchymal transition and they provide a novel perspective on the genesis of the Bochdalek hernia and the evolutionary origin of the diaphragm.

## Results

### The septum transversum, posthepatic mesenchymal plate and pleuroperitoneal folds contain heterogeneous mesenchymal populations

In normal E10.5 embryos, the posterior and dorsal margin of the liver shows an accumulation of mesenchymal tissue, which extends from the dorsal mesenterium of the liver to the lateral tips of the lobes (*Figure 1*). This mesenchymal layer is the PHMP described by *Iritani (1984)*. The PPFs, by this developmental stage, appear as a pair of outgrowths of the body wall located at both sides of the lung buds. (*Figure 1A,B*). They are also constituted of mesenchymal cells lined by the coelomic epithelium. The G2-*Gata4* enhancer directed *LacZ* reporter expression in both PPF and PHMP at E10.5 and E11.5 in the mouse embryo (*Figure 1A*). However, no beta galactosidase activity was observed in the posterior and medial part of the septum transversum, where the liver is connected with the digestive tract (asterisk in *Figure 1A*), indicating that the G2 enhancer is not active in this specific domain.

PHMP and PPFs are not anatomically independent entities. The lateral tips of the PHMP appear continuous with the PPFs at the cephalic level while they are separated more caudally (*Figure 1B*). The continuous part of the PHMP/PPFs ensemble apparently forms itself by the anterior growth of the peritoneal recess, which separates the liver sides from the body wall. Thus, the mesenchymal population of the ST originates the PHMP and also the most cephalic portion of the PPFs. The lateral closure of the pleural cavities occurs between E11.5 and E12.5 by posterior growth of the crescent-

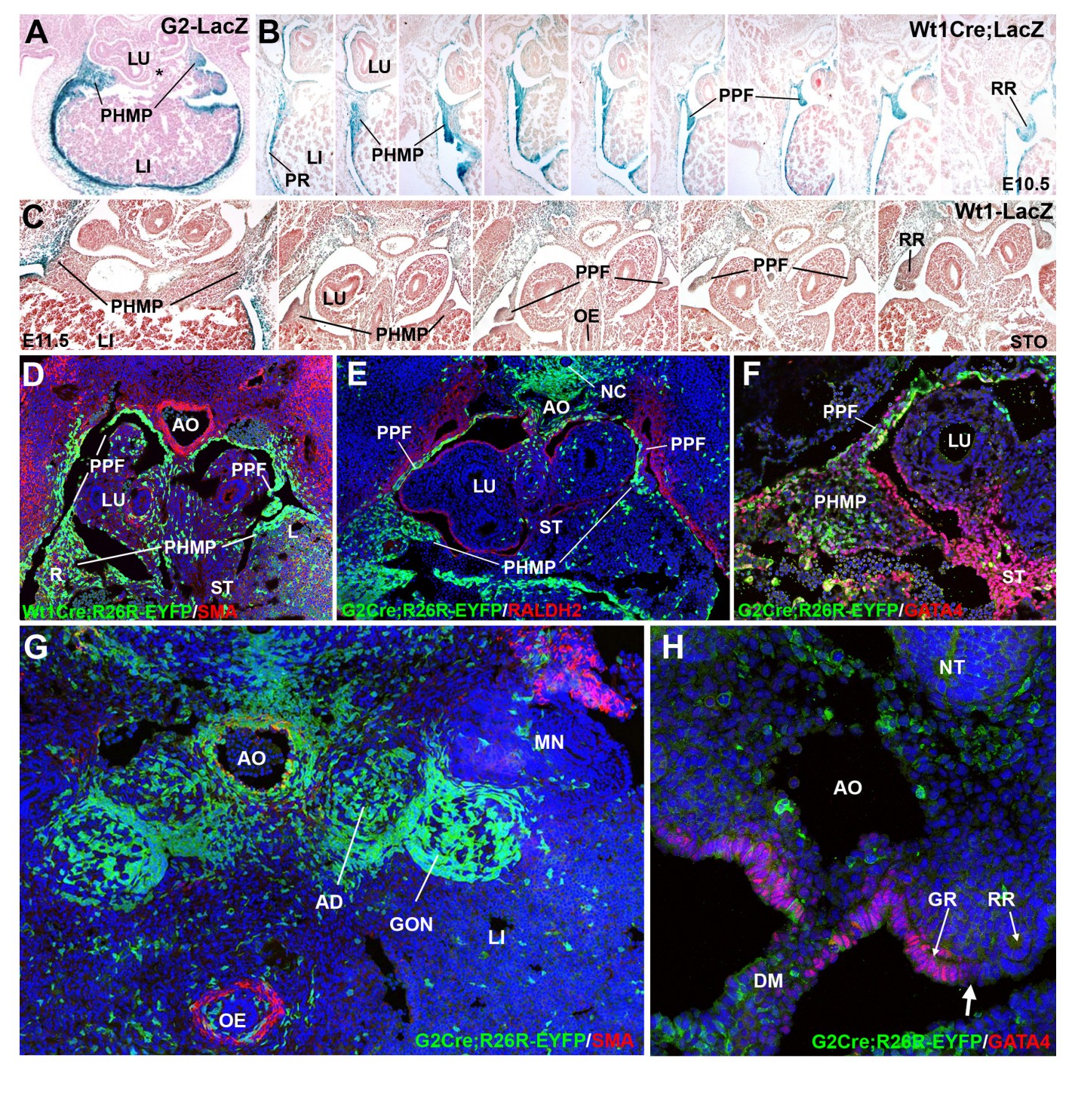

**Figure 1.** Cell lineages in the septum transversum (ST), posthepatic mesenchymal plate (PHMP) and pleuroperitoneal folds (PPF). (**A**) G2-LacZ embryo at the stage E11.5. The G2 enhancer is active in the liver (LI) mesothelium and posthepatic mesenchymal plate (PHMP). However, the central area of the septum transversum shows no activity of this enhancer (asterisk). (**B**) $Wt1^{Cre};R26R^{LacZ}$ embryo, E10.5. Cells from the WT1 lineage appear in blue. Series of transverse sections from cephalic (left) to caudal (right) levels. The anterior peritoneal recess (PR) is splitting the liver from the body wall. The PHMP is continuous with the PPF at cephalic levels. Note the abundance of mesenchymal cells in the anterior PHMP and how this mesenchyme is less abundant in posterior levels. The PPF is continuous with the renal ridge (RR). The mesenchymal cells of all these structures belong to a WT1-expressing cell lineage. (**C**) $Wt1^{LacZ}$ embryo, E11.5. Activation of the $Wt1$ reporter can be seen in the liver mesothelium. PHMP, PPF and RR. The RRs appear at the level of the stomach (STO). OE: oesophagus. (**D**) $Wt1^{Cre};R26R^{YFP}$ embryo, E11.5. Cells from the WT1-expressing cell lineage (green) are more abundant at

*Figure 1 continued on next page*

*Figure 1 continued*

the right PHMP (R) as compared with the left one (L). Note the lack of YFP+ cells in the central area of the ST. AO: aorta; LU: lung. (**E**) G2-*Gata4*$^{Cre}$; *R26R*$^{YFP}$ embryo, E11.5. Cells from the lineage expressing GATA4 under the control of the G2 enhancer are stained in green, and RALDH2 is stained in red. Most cells in the PHMP and PPFs belong to the G2-*Gata4* lineage, but they are very scarce in the central area of the ST. Abundant YFP+ cells are present dorsally to the aorta, and around the notochord (NC). (**F**) G2-*Gata4*$^{Cre}$; *R26R*$^{YFP}$ embryo, E11.5. GATA4 immunostaining in red. Colocalization of GATA4 and YFP is observed in the PHMP and PPF, but the central area of the ST shows GATA4 expression not driven by the G2 enhancer. (**G**) G2-*Gata4*$^{Cre}$; *R26R*$^{YFP}$ embryo, E11.5. Smooth muscle alpha-actin is stained in red. Gonads (GON) and adrenals (AD) contain a large number of G2-*Gata4* lineage cells, but they are scarcer into the mesonephros (MN). Data reused, with permission, from Figure 4A, Muñoz-Chápuli et al. Developmental Dynamics, Special Issue: Mechanisms of Morphogenesis, 245:307–322 (2016). © 2015 Wiley Periodicals, Inc. (**H**) Immunolocalization of GATA4 (red) in an E10.5 G2-*Gata4*$^{Cre}$; *R26R*$^{YFP}$ embryo. The G2 lineage is shown in green. There is a sharp boundary (arrow) between the mesothelial cells expressing GATA4 of the genital ridge (GR), close to the dorsal mesentery (DM), and the mesothelial cells of the renal ridges (RR), which are GATA4-. AO: aorta; OE: oesophagus.

like continuum formed by the PHMP and the PPFs, a process which is parallel to the fast growth of the liver lobes during these stages (*Figure 1B,C*).

As seen in the *Wt1*$^{Cre}$ labeled cells, the PPFs continue posteriorly as the nephric ridges, the progenitor tissue of the mesonephros (*Figure 1B and C*). A loose mesenchymal tissue appears in the PPF between the more compact arrangement of the cells in the PHMP and in the renal ridges, but the commissural area of the PPF is always constituted by compact mesenchyme similar to that from the PHMP (arrow in *Figure 2C*). All these mesenchymal populations are derived from a WT1-expressing cell lineage, as shown by two models, the *Wt1*$^{Cre}$;*R26R*$^{LacZ}$ (*Figure 1B*) and the *Wt1*$^{Cre}$; *R26R*$^{YFP}$ (*Figure 1D*). Expression of the *Wt1* gene is also revealed in the PPF and PHMP mesenchyme by the WT1-LacZ reporter and by immunohistochemistry (*Figures 1C* and *3A*, respectively).

Epithelial and mesenchymal cells of the PHMP and most anterior PPFs derive from a cell lineage in which *Gata4* expression is driven by the G2 enhancer. This is demonstrated by the expression of YFP in G2-*Gata4*$^{Cre}$;*R26R*$^{YFP}$ embryos, and supports the existence of a process of cell migration from PHMP to PPFs (*Figure 1E,F*). However, the nephric ridges and the mesonephros derived from them show no YFP+ cells in this model, indicating that the progenitors of these tissues derived from a different cell lineage. Thus, G2-*Gata4* lineage tracing allows to establishing a well-defined limit between two lateral mesodermal territories that we will call herein the G2$^{+}$ and the G2$^{-}$ domains (see *figure 3B,C*). Interestingly, the gonad and the adrenal stroma were YFP+ (*Figure 1G*), and this can be related with the early expression of GATA4 in the coelomic epithelium, which will cover the genital but not the renal ridges (*Figure 1H*). Incidentally, the GATA4 positive cells from the central part of the ST do not express YFP in G2-*Gata4*$^{Cre}$;*R26R*$^{YFP}$embryos (*Figure 1F*), an observation consistent with the lack of reporter activity in the G2- *Gata4*$^{LacZ}$ embryos (*Figure 1A*). These data reveal a significant heterogeneity between central and lateral mesenchymal populations in the ST.

A clear asymmetry was observed in the distribution of PHMP mesenchyme in embryos of E10.5-E12.5 (*Figure 1D,E*). Mesenchymal cells were always more abundant in the right PHMP than in the left, which might favor the herniation in the left side. We estimated, by image analysis, the volume taken up by the YFP+ cells in the right and left PHMP of four G2-*Gata4*$^{Cre}$; *R26R*$^{YFP}$ embryos. The R/L ratio of these volumes was 1,21 and 1,45 in two E11.5 embryos, and 1,53 and 1,61 in two E12,5 embryos (mean = 1,45; S.E.M = 0,086). The t-test for one sample compared with an expected value = 1 (symmetrical distribution) gave a result of 5,21 (p-value=0,0069 for three d.o.f). Thus, the right PHMP contains about 50% more cells from the G2-*Gata4* lineage than the left one.

## WT1 conditional knockout mice display Bochdalek's hernia

This is a basically descriptive study of the phenotype of mouse embryos with conditional deletion of Wt1 in lateral mesoderm. We have included in the study 36 out 104 mutant embryos detected by genotyping. The rest of them had been previously used for a study of the cardiac phenotype (*Cano et al., 2016*).

To understand the role of WT1 positive cells in the development of diaphragm, we conditionally inactivated *Wt1* gene using the described G2-*Gata4*$^{Cre}$ line. Deletion of WT1 in the PHMP/PPFs continuum, i.e. in the G2$^{+}$ domain, causes a phenotype characterized by defects in the inflow tract of the heart, coronary vessels (*Cano et al., 2016*) and, importantly, Bochdalek's hernia, most frequently

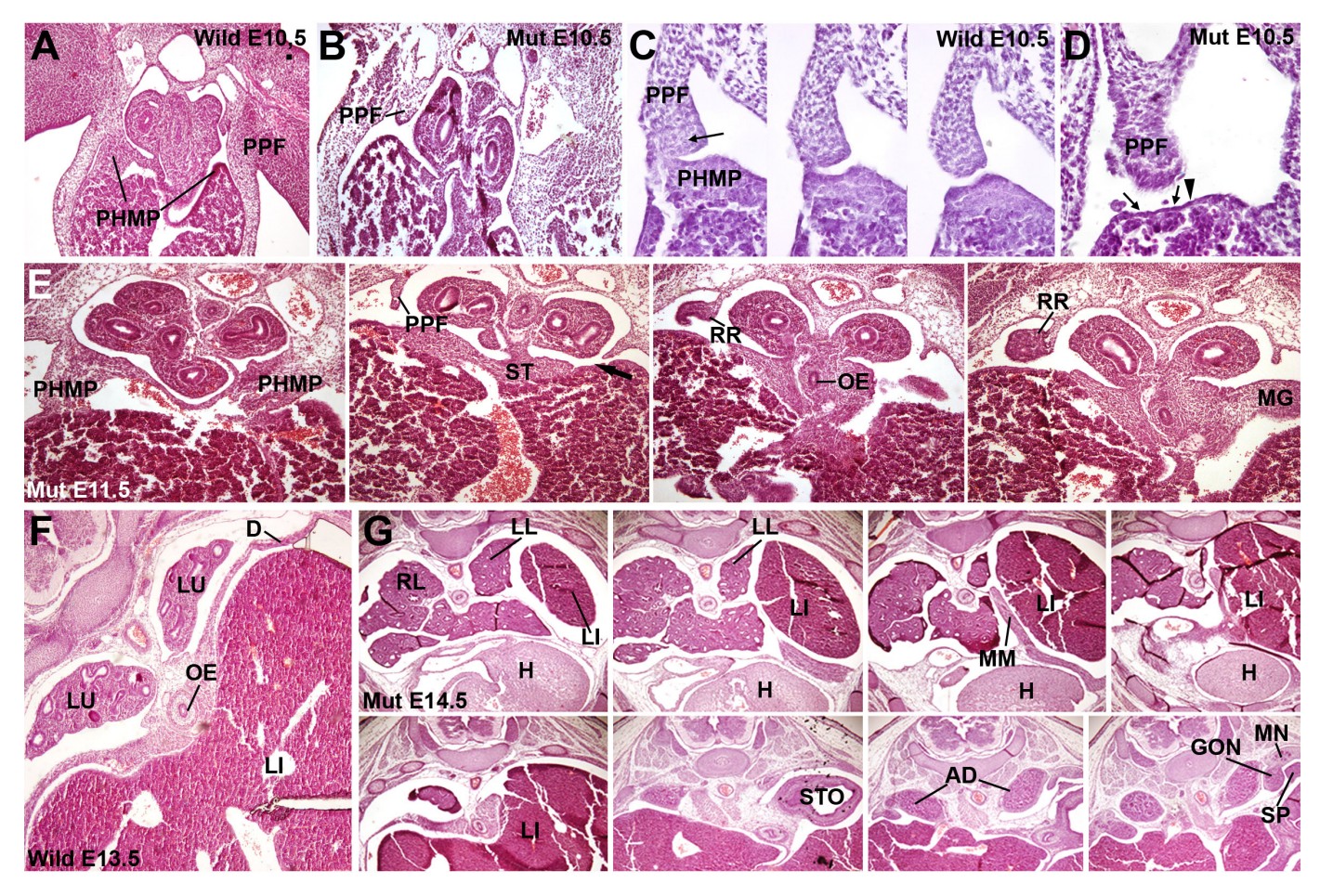

**Figure 2.** Phenotype of the G2-*Gata4*^*Cre*; *Wt1*^*fl/fl* embryos. (**A–D**) Wildtype (**A,C**) and mutant (**B,D**), E10.5 littermates. The larger amount of mesenchymal cells in the posthepatic mesenchymal plate (PHMP) is evident in the wildtype, especially in the right side. Serial sections corresponding to the connection between the PHMP and the pleuroperitoneal folds (PPF) are shown in **C**. Note the presence of compact mesenchyme in the PHMP and also in the closest part of the PPF (arrow in **C**). A corresponding section of the mutant in shown in **D**. Note the lack of PHMP, the coelomic epithelium lying directly on the hepatic tissue (arrows in **D**) and the limit of the septum transversum (ST) mesenchymal cells (arrowhead in **D**), which do not extend laterally. (**E**) Serial sections of a G2-*Gata4*^*Cre*; *Wt1*^*fl/fl* E11.5 embryo at levels equivalent to those shown in *Figure 1C*. Despite the presence of normal PHMP in the anterior part, the posterior areas of the liver lack of lateral mesenchymal cells (arrow). The mesenchyme is restricted to the central ST. Renal ridges (RR) appear at a level corresponding to the entrance of the oesophagus (OE) into the ST. MG: mesogastrium. (**F**) Wildtype E13.5 embryo showing complete isolation of the pleural cavities by the pleuroperitoneal membranes that constitute the main part of the diaphragm (D). (**G**) G2-*Gata4*^*Cre*; *Wt1*^*fl/fl* E14.5 embryo at eight different levels showing left diaphragmatic defect with herniation of the left liver lobe (LI) into the pleural cavity and severe hypoplasia of the left lung (LL). Ectopic muscle appear in the mediastinum (MM). Adrenals (AD), mesonephros (MN), gonads (GON) and spleen (SP) appear normal. LU: lungs; H: heart; RL: right lung; STO: stomach.

located in the left side (*Figure 2*). The genotypes of the embryos obtained at different ages show the expected percentage of mutants at all embryonic stages (about 25%) except for the stage E15.5, when we only obtained four mutants among 60 embryos (*Table 1*). The number of embryos studied in older stages (42 embryos, nine mutants, 21,4%), was not enough to confirm if the mutation causes some lethality in late gestation.

We analyzed WT1 mutant embryos at different stages of development. Defect in the PHMP in G2-*Gata4*^*Cre*;*Wt1*^*fl/fl* mutant embryos could be observed as early as E10.5, since less mesenchymal cells are present between the coelomic epithelium and the hepatoblasts when compared with control littermates (*Figure 2A–D*). At later stages, the closure of the pleural cavity by growth of the ST/PPFs crescent is delayed with respect to the controls, and the mesenchymal cells of the PHMP are

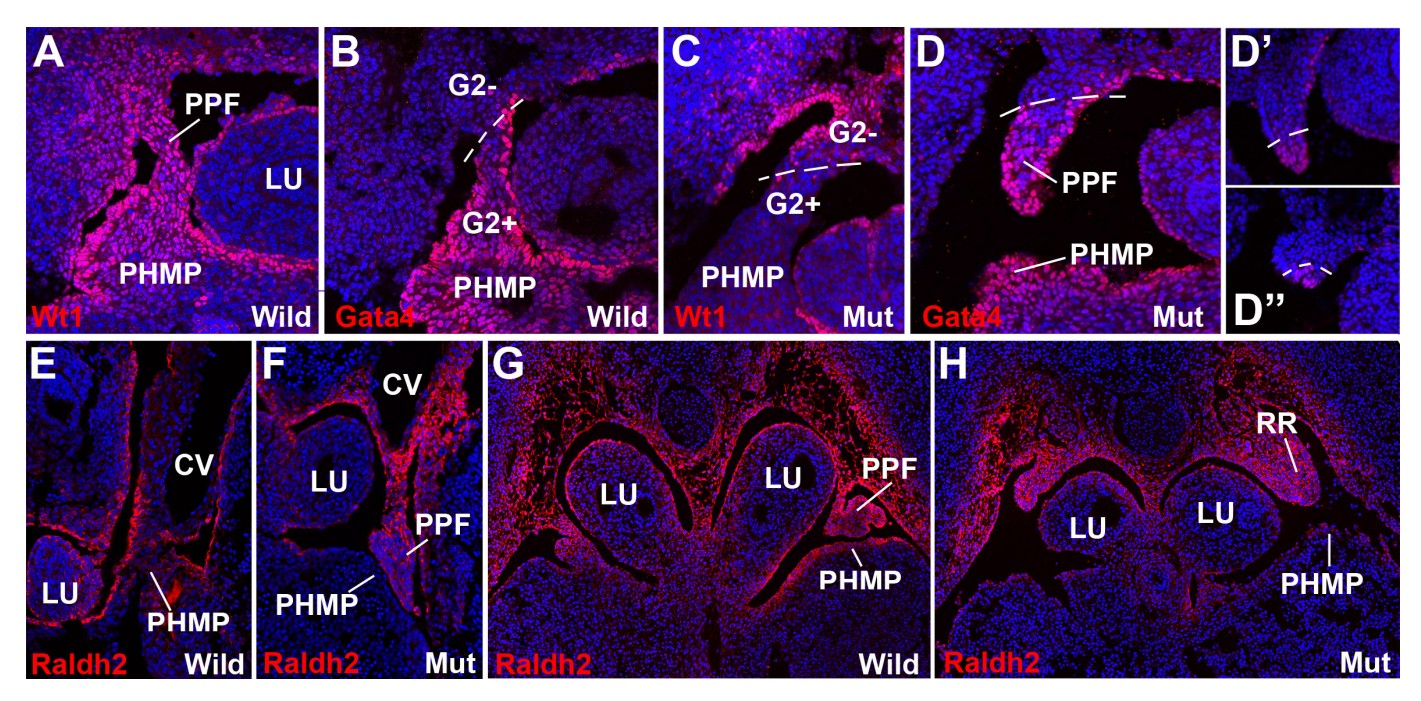

**Figure 3.** WT1, GATA4 and RALDH2 expression in G2-*Gata4^Cre*; *Wt1^fl/fl^* embryos. (**A,B**) Immunolocalization of WT1 (**A**) and GATA4 (**B**) in wildtype E11.5 embryos. The posthepatic mesenchymal plate (PHMP) and pleuroperitoneal folds (PPF) show abundant positive cells. However, the WT1 immunoreactive cells are present in the base of the PPF and the adjacent body wall, where GATA4+ cells are absent. This difference defines the G2+ and the G2- domains. Data in A reused, with permission, from Figure 3E, Muñoz-Chápuli et al. Developmental Dynamics, Special Issue: Mechanisms of Morphogenesis, 245:307–322 (2016). © 2015 Wiley Periodicals, Inc. (**C–D**) Immunolocalization of WT1 (**C**) and GATA4 (**D**) in G2-*Gata4^Cre*; *Wt1^fl/fl^* E11.5 embryos. The PHMP and the G2+ domain of the PPF show GATA4+ cells but no WT1+ cells due to the conditional deletion of this gene in the G2+ domain. WT1 expression remains in the G2- domain of the base of the PPF and body wall. GATA4 expression is normal in most cephalic PPF, but immunoreactivity disappears in more caudal areas of PPF, as shown in **D'** and **D''**. (**E–H**) Immunolocalization of RALDH2 in E10.5 (**E,F**) and E11.5 (**G,H**) wildtype and G2-*Gata4^Cre*; *Wt1^fl/fl^* mutant embryos. RALDH2 immunoreactivity is high in PPF and the adjacent body walls. Note strong immunoreactivity in the renal ridges (RR) of the E11.5 mutant embryo. However, RALDH2 immunoreactivity is reduced or absent in the PHMP and liver mesothelium of the mutant embryos, as compared with the controls. CV: cardinal veins; LU: lungs.

located more medially, not in the lateral tips of the liver lobes. This is more evident at the left side (arrow in *Figure 2E*). Thus, these cells cannot migrate towards the PPFs precluding caudal growth of the lateral commissures of the pleuroperitoneal septa and closure of the pleural cavities.

Five E12.5 mutant embryos studied already showed strong reduction of the PHMP size as compared with control littermates, but the diaphragmatic defect of the WT1 conditional knockout embryos became clearly apparent by E13.5, when the pleural cavities are almost completely isolated in wildtype embryos (*Figure 2F*). Of 23 mutant embryos analyzed by the stages E13.5 or older, only four of them showed normal development of the diaphragm, while the rest of embryos showed different degrees of diaphragmatic defects (*Table 1*). Three of the embryos showed thin, membranous diaphragm in the left side and 16 showed a wide defect in the diaphragm, frequently with herniation of the liver into the left pleural cavity and hypoplasia of the left lung (*Figure 2G*). One of them (E13.5) showed the defect only at the right side and the CDH was present in the left side in the rest of them. Thus, about 80% of the E13.5 or older mutant embryos showed abnormal diaphragmatic development.

Renal ridges appear well developed in mutant E11.5 embryos at a level corresponding to the entrance of the oesophagus into the ST (*Figure 2E*). The presence of intermediate mesoderm in these thoracic renal ridges was confirmed by ISH of Pax2 (*Figure 4*). In wildtype embryos of the same age thoracic renal ridges can also be seen, but located more caudally, at the level of the stomach (*Figure 1C*).

**Table 1.** Number of embryos studied and % of G2-*Gata4*<sup>Cre</sup>;*Wt1*<sup>fl/fl</sup> found. The five E12.5 embryos showed abnormal posthepatic mesenchymal plates at the left side, but they were not computed for the diagnosis of the diaphragmatic hernia, that was performed only on the 23 embryos at the stages E13.5 or older.

| Stage | Number of embryos studied | Number and% G2Cre+; Wt1fl/fl | Embryos analyzed for phenotype | No defect | Mild defect | Diaphragmatic hernia |
|---|---|---|---|---|---|---|
| E10–10.5 | 42 | 11 (26,2%) | 3 | | | |
| E11.5 | 89 | 21 (23,6%) | 5 | | | |
| E12.5 | 76 | 21 (27,6%) | 5 | | | (5) |
| E13,5 | 65 | 19 (27,1%) | 13 | 1 | 1 | 11* |
| E14,5 | 53 | 17 (32,1%) | 4 | 2 | | 2 |
| E15,5 | 60 | 4 (6,7%) | 4 | | 2 | 2 |
| E16,5 | 21 | 4 (19,0%) | 1 | 1 | | |
| E17,5–18,5 | 21 | 5 (23,4%) | 1 | | | 1 |
| Total | 427 | 102 (23,6%) | 36 | 4 (17,4%) | 3 (13,0%) | 16 (69,6%) |

*We have included seven embryos from females fed with control diet in the RA-rescue experiment. One embryo showed CDH only at the right side.

The sharp boundary between the above explained G2$^+$ and G2$^-$ domains was clearly observed by the immunolocalization of WT1 and GATA4 protein in G2-*Gata4*<sup>Cre</sup>;*Wt1*<sup>fl/fl</sup> mutant embryos (*Figure 3*). WT1 immunoreactivity is strong in the PHMP and PPF of the wildtype embryos and localizes in the coelomic epithelium and, with a lower intensity, in mesenchymal cells (*Figure 3A*). However, WT1 is clearly downregulated in the G2$^+$ domain of the mutant embryos indicating an efficient

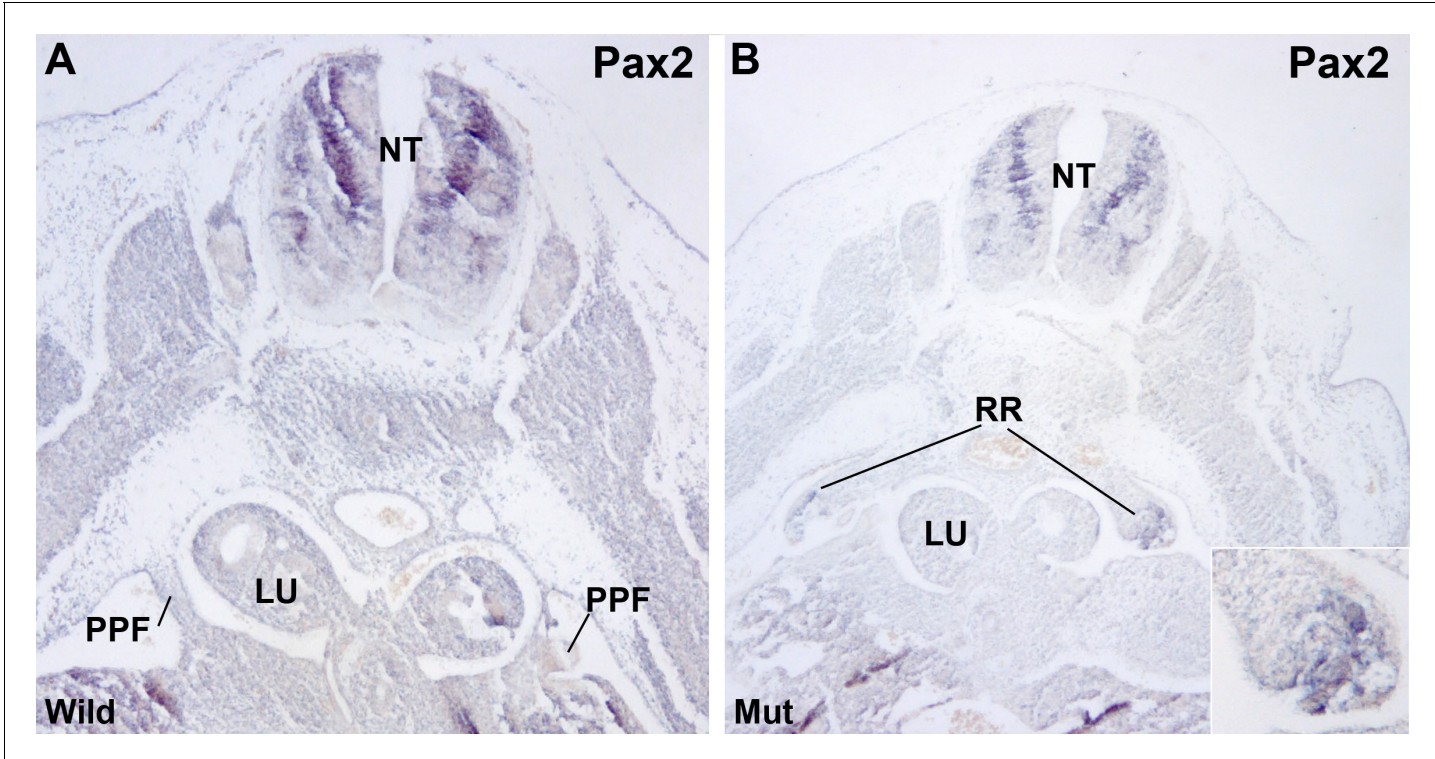

**Figure 4.** Localization of Pax2 expression by in situ hybridization. (**A**) Control E11.5 embryo. Pax2 is expressed in the neural tube (NT). The pleuroperitoneal folds (PPF) lack of Pax2 expression. (**B**) G2-*Gata4*<sup>Cre</sup>; *Wt1*<sup>fl/fl</sup> E11.5 embryo. Pax2 is expressed in the PPF at the level of the lung buds (LU). The expression is stronger in the tubular structures that are developing in the left PPF (insert).

excision of the WT1 floxed allele (*Figure 3C*). Immunolocalization of GATA4 protein marked the border between both G2 domains, confirming the absence of endogenous GATA4 protein and the inactivity of the G2 enhancer in the G2⁻ domain. (*Figure 3B,D*). The presence of GATA4+ cells in the more cephalic part of the PPF supports a migration of PHMP cells towards the PPFs.

Since WT1 is involved in epithelial-mesenchymal transition (EMT) of the epicardium by repression of E-cadherin and activation of Snail1 (*Martínez-Estrada et al., 2010*), we checked the expression of E-cadherin in the developing PHMP of G2-*Gata4*^*Cre*^;*Wt1*^*fl/fl*^ mutant embryos. Control E11.5 embryos show an E-cadherin negative coelomic epithelium over a layer of mesenchymal cells in the developing PHMP (*Figure 5A–D*). However, mutant embryos show expression of E-cadherin in the epithelium of the corresponding area and lack of E-cadherin negative mesenchymal cells (*Figure 5E–H*).

## Systemic WT1 loss of function leads to complete lack of PHMP

We have compared the G2-*Gata4*^*Cre*^;*Wt1*^*fl/fl*^ CDH phenotype with that displayed by the systemic WT1 knockout mouse embryos, which do not survive beyond E13.5. Despite the early embryonic death of these models, we have observed a more severe defect, with a complete lack of PHMP and displaying, in some cases, persistent thoracic nephric ridges in E12.5 and E13.5 embryos (*Figure 6*). In these embryos only the central part of the ST is conserved, i.e. the area where the G2 enhancer is not active, as shown in *Figure 1E*.

## Retinoic acid partially rescues the defects in the diaphragm of WT1 conditional knockout embryos

The retinoic acid (RA) signaling pathway has been involved in the formation of diaphragm. To test whether RA signaling might be disturbed in our CDH model, we checked the expression of RALDH2 by immunofluorescence in G2-*Gata4*^*Cre*^; *Wt1*^*fl/fl*^ mutant and control embryos. RALDH2 expression is found in the PHMP and PPF of wildtype E10.5-E11.5 embryos (*Figure 3E,G*). The PPF of the G2-

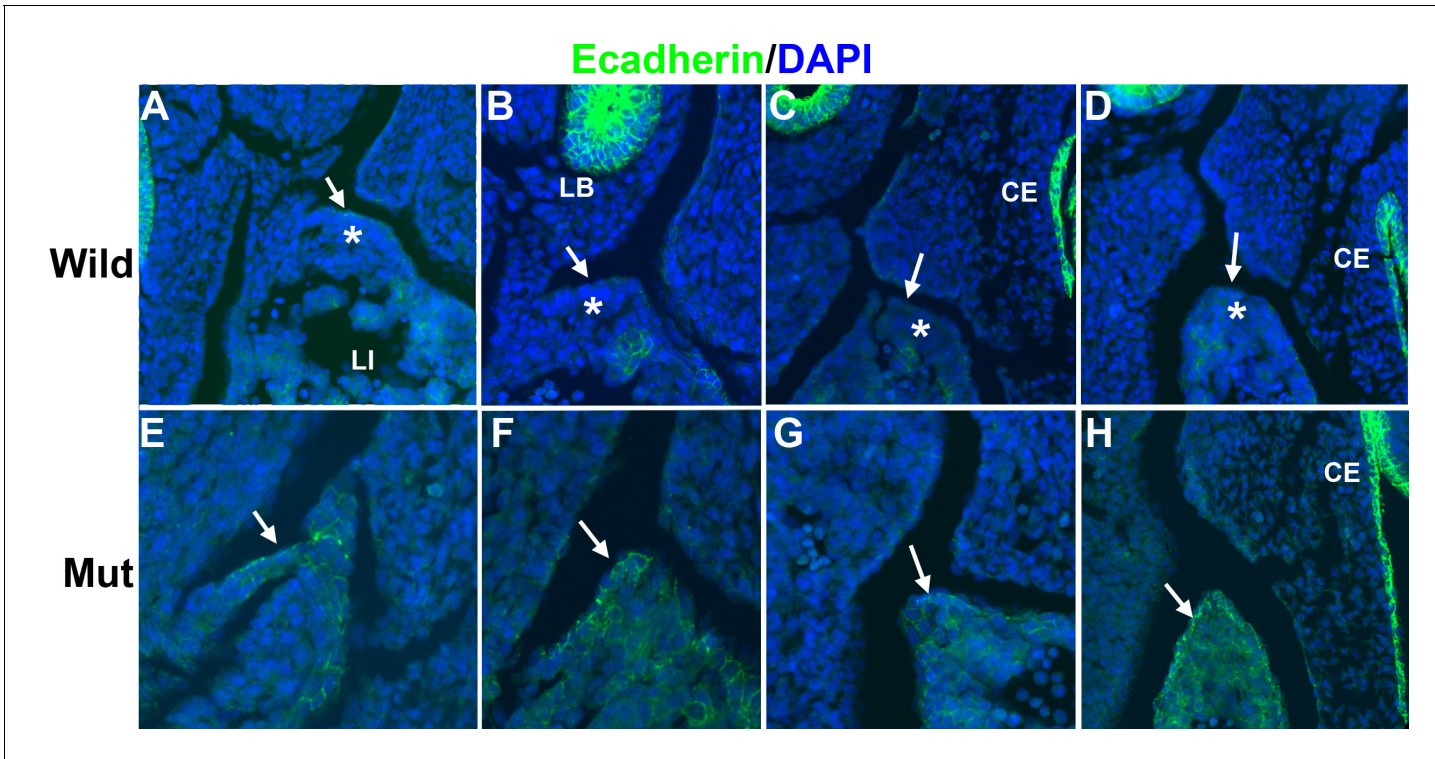

**Figure 5.** Immunolocalization of E-cadherin in the left posthepatic mesenchymal plate (PHMP) of four E11.5 control embryos (**A–D**) and four E11.5 G2-*Gata4*^*Cre*^; *Wt1*^*fl/fl*^ embryos (**E–H**). E-cadherin appears upregulated in the coelomic epithelium of the PHMP of the mutant embryos (arrows in **E–H**). In wildtype embryos the coelomic epithelium of the PHMP lacks of E-cadherin expression (arrows in **A–D**) and covers a layer of E-cadherin-negative mesenchymal cells (asterisks).

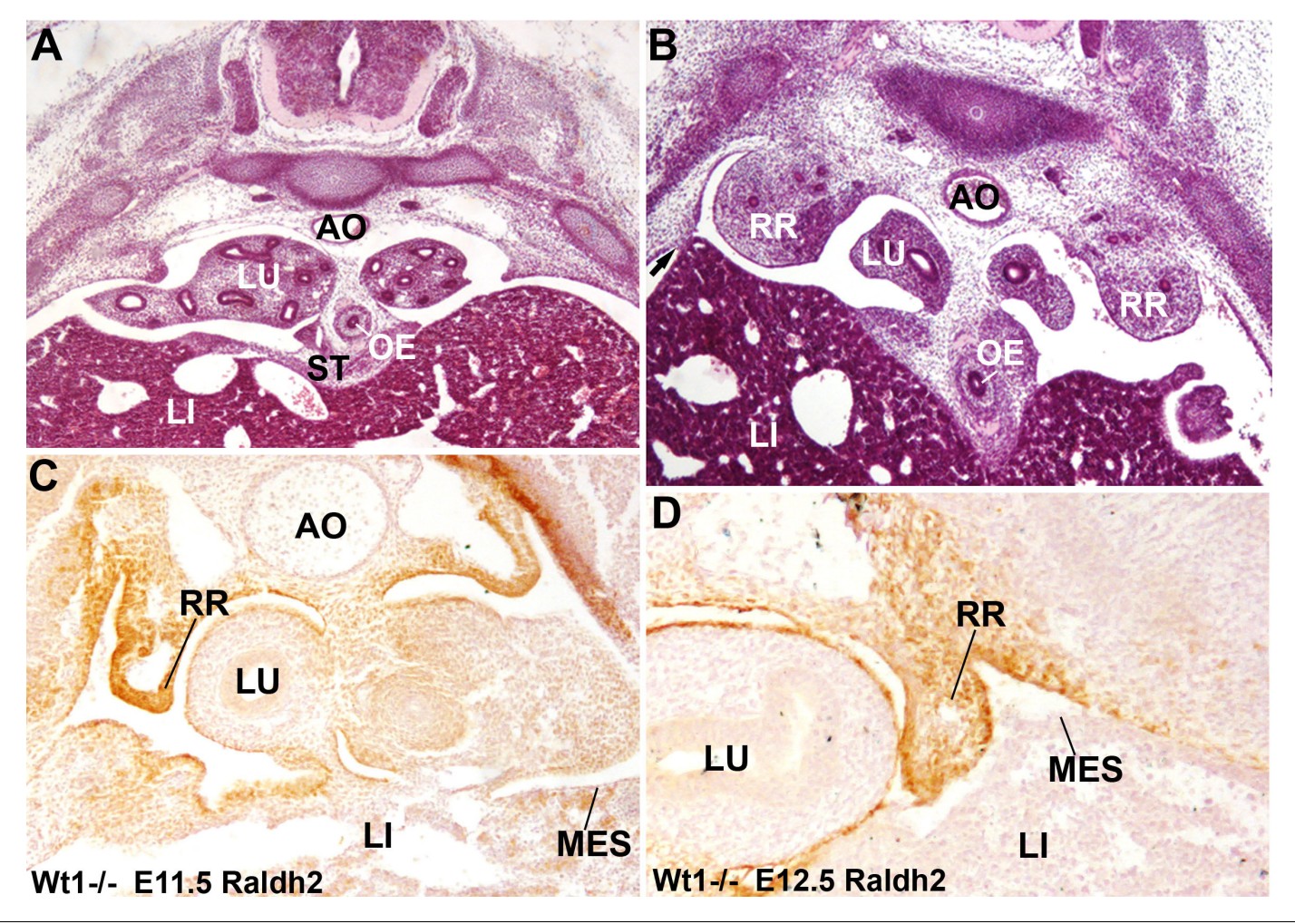

**Figure 6.** Diaphragmatic phenotype in systemic WT1 loss of function. (A,B) WT1-/- E13.5 embryos. The complete lack of posthepatic mesenchymal plate is evident, mesenchymal cells are absent from the lateral ends of the liver lobes and remain restricted to the central area of the septum transversum (ST), close to the oesophagus (OE). The pleural cavities are continuous with the peritoneal cavity. Note the renal ridges (RR) persisting at thoracic level, and the abnormal adhesion of the right liver lobe with the body wall (arrow in **B**). AO: aorta; LI: liver; LU: lungs. (C,D) Immunolocalization of RALDH2 in WT1-/- embryos, E11.5 (**A**) and E12.5 (**B**). RALDH2 immunoreactivity is strong in the renal ridges (RR), but weak or absent in the liver (LI) mesothelium (MES). AO: aorta; LU: lung.

*Gata4^Cre^; Wt1^fl/fl^* mutant embryos show an increased immunoreactivity for RALDH2 but we found a reduced or absent expression of this enzyme in the PHMP (*Figure 3F,H*). This probably reflects, as discussed below, the persistence of the intermediate mesoderm of the nephric ridges into the PPF. In fact, RALDH2 is normally expressed at high levels in the intermediate mesoderm even in systemic WT1 knockout embryos, which also show decreased RALDH2 immunoreactivity in the liver mesothelium (*Figure 6C,D*).

RA treatment partially rescues the pulmonary phenotype and promotes alveologenesis in the nitrofen model of induced CDH in rats (*Montedonico et al., 2008*; *Sugimoto et al., 2008*). In order to check the effects of a RA treatment on our mutant phenotype, in a first experiment we fed pregnant females with a RA-supplemented diet. For ethical reasons we decided to limit the study to four litters in which we detected eight G2-*Gata4^Cre^; Wt1^fl/fl^* mutant embryos. Five of them (62,5%) displayed a complete septation of the pleural cavity by E15.5, i.e. about three times the proportion of normal phenotypes found in the non-treated mutants (Z test = 2,09, two-tailed p-value= 0,037). Control littermates were not affected by the RA-treatment.

Then, we designed a second experiment to assess the effect of the RA dietary supplement on the development of the diaphragmatic defect, comparing E13.5 mutant embryos with control embryos of the same age obtained from pregnant females fed with the same diet without the retinoic acid supplement (*Figure 7*). Seven mutant embryos from three different females fed with control chow showed CDH (*Figure 7B*). For ethical reasons we decided to limit the study to four litters from females fed with RA-supplemented diet, where we identified nine mutant embryos. One of them showed normal diaphragm, three displayed CDH and five showed a small opening on the most posterior angle of the left pleural cavity (*Figure 7C*). Interestingly, two out of five control littermates also showed a similar opening in the same area (*Figure 7A*). Thus, we measured on digitalized images the width of the left diaphragmatic discontinuity in the eight non-treated mutants, the seven RA-treated mutants and the two RA-treated controls displaying this opening. The size of the diaphragmatic discontinuity was significantly smaller in the RA-treated mutants as compared with the control mutants (Student's t test, p-value=0,009), but it was no significantly different when compared with the defect of the two RA-treated controls (Student's t test, p-value=0,20) (*Figure 7D*).

## Discussion

Classical descriptions of the origin of the diaphragm state that this structure starts its development by fusion of the dorsal PPFs with the ST. Later, the liver partially detaches from this ST/PPF membrane, which becomes muscularized by migration of myoblasts from the somite. Thus, in the classical scenario CDH can arise either by a defect in the ST/PPF fusion process or by a failure of myoblasts migration. However, very few descriptions take into account that PPFs and ST are not anatomically independent structures. Along the dorsal body cavity of the mammalian embryo a pair of long crests are located at both sides of the dorsal mesentery. In the posterior part, these crests form the nephric ridges. In the anterior end of the peritoneal cavity they form the PPFs, which descend to a more lateral level and directly merge with the ST (where the liver is expanding) through the so called PHMP (*Iritani, 1984*; *Hayashi et al., 2011*). This continuity between PHMP, PPFs and nephric ridges is beautifully shown by *Mayer et al. (2011)* through scanning electron microscopy. The boundary between PPFs and PHMP is diffuse although the trajectory of the phrenic nerve has been proposed as a limit between these territories (*Babiuk, 2003*). However, our study of the distribution of the YFP + cells reveals a novel and well-defined boundary between the lateral plate (G2$^+$) and the intermediate (G2$^-$) mesodermal domains. The lateral mesoderm domain would be characterized by the expression of *Gata4* driven by the enhancer G2 while the intermediate mesoderm domain would lack *Gata4* expression. The limit between G2$^+$ and G2$^-$ domains is established as early as E10.5 in the coelomic epithelium lateral to the mesentery (*Figure 1H*), between the genital and the nephric ridges. A second boundary appears between the PHMP and the central area of the posterior ST, which shows strong GATA4 expression not driven by the G2 enhancer. This heterogeneity between ST mesenchymal populations is relevant since the central mesenchyme of the ST will not be involved in diaphragm development, as demonstrated by *Merrell et al. (2015)* and discussed below.

We think that another important factor in order to understand the development of the CDH is the asymmetrical distribution of the mesenchyme in the PHMP/PPFs continuum. Asymmetry has already been described in the gut mesenchymal compartment, where cells are more densely packed on the left than on the right (*Kurpios et al., 2008*). However, differences in the amount of mesenchymal cells in the PHMP or ST had not been reported, although *Mayer et al. (2011)* pointed to a larger surface contact between PPFs and PHMP at the right side. We have shown that the right part of the PHMP shows more mesenchymal cells than the left part by the time in which the pleuroperitoneal septa develop, and we believe that this fact is relevant for the left predominance (>85%) of the Bochdalek CDH.

Our results suggest that CDH could be caused by a defective population of the PHMP/PPF continuum by mesenchymal cells. A process of migration of liver mesenchymal cells into the PPFs relevant for the closure of the pleuroperitoneal canal had been suggested in human embryos (*Hayashi et al., 2011*). Since this closure should be dependent on the amount of cells, the smaller amount of mesenchyme observed in the left side would make it more difficult to complete the process. Thus, we think that left predominance of Bochdalek hernia can be connected with primary mechanisms of left-right visceral asymmetry, and particularly with an asymmetrical generation of mesenchyme.

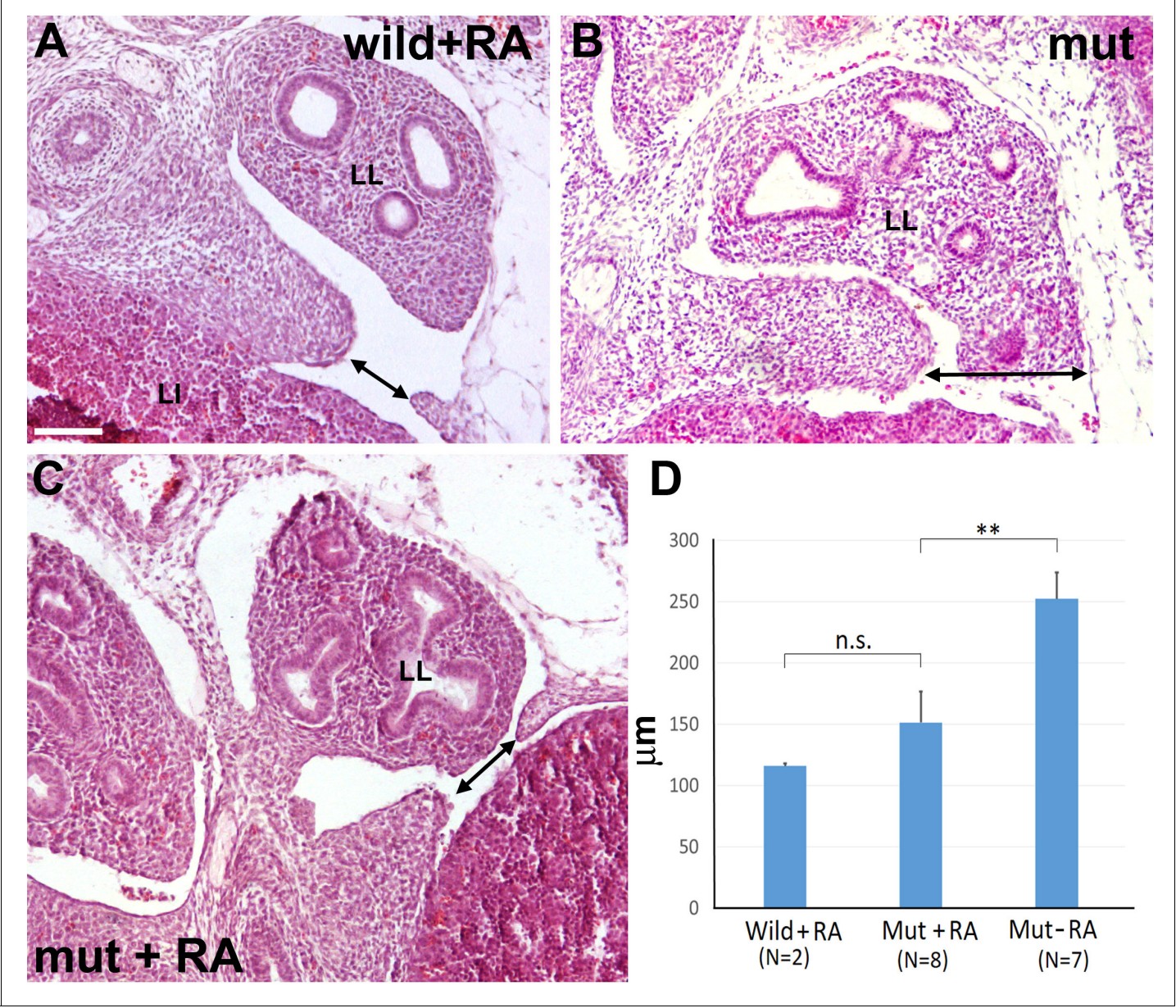

**Figure 7.** G2-*Gata4*<sup>Cre</sup>; *Wt1*<sup>fl/fl</sup> mutant embryos and wildtype littermates collected at stage E13.5 from females fed with a retinoic acid (RA) supplemented diet or with control diet by days E8.5-E10.5 of pregnancy. (**A**) Wildtype embryo treated with RA. A small discontinuity appears in the left pleuroperitoneal septum (arrow). (**B**) Mutant embryo not treated with RA. A wide defect is evident in the left pleuroperitoneal septum. (**C**) Mutant embryos treated with RA and showing a discontinuity similar to that described in **A**. (**D**) Width of the discontinuity of the pleuroperitoneal opening in mutant and control embryos treated with RA compared with mutant embryos not treated with RA. The latter embryos show significantly wider openings than the mutant embryos treated with RA. LI: liver; LL: left lung. Scale bar = 100 μm.

The following source data is available for figure 7:

**Source data 1.** Measurements of the pleuropericardial opening used for *Figure 7D*.

As we have shown, WT1 is required in the G2$^+$ domain (PHMP and anterior part of the PPFs) for the formation of the pleuroperitoneal septa. The immediate consequence of its deletion in this territory is a lesser number of mesenchymal cells in the PHMP. Furthermore, WT1-null embryos completely lack of PHMP. WT1 is critically involved for EMT of the epicardium (*Martinez-Estrada et al., 2010*). Thus, a hypothetical function of WT1 in the PHMP/PPF could be to promote

EMT and generation of mesenchyme. The upregulation of E-cadherin concomitant with the lack of mesenchymal cells in the PHMP of the mutant embryos supports this suggestion. Furthermore, WT1 plays the same role in the liver mesothelium (*Ijpenberg et al., 2007*), and a recent report on the conditional deletion of β-catenin in the WT1 lineage leading to CHD (*Paris et al., 2015*) stresses again the relationship between WT1 and EMT. Significantly, the loss of function of β-catenin phenocopies the loss of function of WT1 when its deletion is induced between E10.5 and E11.5, the time window in which we suggest that WT1 is promoting EMT in the coelomic epithelium of the PHMP.

The coelomic mesothelial origin of human diaphragmatic malformations, and their relationship with WT1 function were proposed by *Suri et al. (2007)* in a study on patients with Meacham syndrome, a rare multiple malformation affecting to genitalia, heart and diaphragm. Two out of eight patients with this syndrome had heterozygous mutations in the zinc finger domains of WT1. Thus, our G2-*Gata4*$^{Cre}$;*Wt1*$^{fl/fl}$ mice, displaying congenital cardiac (*Cano et al., 2016*) and diaphragmatic anomalies, can be a valuable animal model for this kind of human pathology.

The presence of lateral plate mesoderm into the PPFs coincides with the backward movement of the intermediate mesoderm from the thoracic levels. Significantly, *Volpe et al. (2008)* report increase of *Hoxb4* expression in the nitrofen model of CDH. This is relevant since *Hoxb4* is a key gene to determine the nephric fate of the intermediate mesoderm. In fact, ectopic expression of *Hoxb4* in anterior non-kidney mesoderm, either by retinoic acid (RA) administration or plasmid-mediated overexpression, resulted in ectopic kidney gene expression (*Preger-Ben Noon et al., 2009*). This is paradoxical, since nitrofen is reducing RA signaling, and *HoxB4* is induced by this morphogen. Our explanation for this paradox is that nitrofen would preclude migration of PHMP mesenchyme towards the PPFs, leaving the intermediate mesoderm (which normally expresses HoxB4) in its primitive anterior position.

WT1 has a dual function concerning the epithelial-mesenchymal transition and the reverse process, mesenchymal-epithelial transition (MET). The first process is promoted by WT1 in heart and liver (G2+ domain), but MET is induced by WT1 in the kidneys (G2- domain) (*Essafi et al., 2011*) that lack a significant coelomic contribution (*Ariza et al., 2016*). We think that there is a connection between these observations. GATA4 expression seems to be independent of WT1 (*Ijpenberg et al., 2007*; *Delgado et al., 2014*). The involvement of GATA4 in CDH is probably related with the differentiation of the PHMP/PPFs mesenchyme and the migration of myoblasts (*Merrell et al., 2015*) during the late stages of diaphragm formation. Instead, WT1 would be involved in an earlier developmental stage, promoting the local generation of the PHMP/PPFs mesenchyme. This explains differences between the *Prrx1*$^{Cre}$;*Gata4*$^{fl/fl}$ model of Merrell et al. (bilateral, with membranous sac and defects of muscularization) and our WT1 model, much more similar to the human Bochdalek hernia (left-sided and with discontinuity of the pleuroperitoneal septum). Incidentally, we think that the claim made by *Merrell et al. (2015)* about the lack of contribution of the ST to the diaphragm should apply only to the central area of the ST where GATA4 expression is not driven by the G2 enhancer, as shown by us.

In summary, two types of CDH would originate in different moments of development. CDH due to defective generation of the PHMP mesenchyme appears in models of RA deficiency and also in models with loss of function of WT1. In later stages CDH can develop by defective muscularization of the pleuroperitoneal septa. This type of CDH occurs in models of GATA4 conditional deletion (*Merrell et al., 2015*) and also in c-Met deficiency (*Babiuk and Greer, 2002*).

The dietary supplement of RA to pregnant females that rescues pulmonary hypoplasia in the nitrofen model of CDH (*Montedonico et al., 2008*; *Sugimoto et al., 2008*) was also partially effective in our model of conditional deletion of WT1. The proportion of E15.5 normal mutant embryos increased from 21% to 62.5% after this treatment. In E13.5 embryos, RA treatment significantly reduced the size of the left pleural cavity opening. RALDH2 immunoreactivity decreases or disappears in the PHMP of mutant embryos, and this expression is probably activated by WT1 as it occurs in the epicardium (*Guadix et al., 2006*), where RA signaling is essential for epicardial EMT (*von Gise et al., 2011*).

The continuity between septum transversum, PPFs and nephric ridges suggests an evolutionary scenario about the origin of the diaphragm in mammals. The diaphragm probably derives from the cephalic end of the nephric ridges, which run throughout the peritoneal cavity in fish, amphibians and reptiles. We suggest that the thoracic nephric ridges were invaded in the ancestors of mammals by lateral plate mesodermal mesenchyme, giving rise to the pleuroperitoneal septa. In fact, the idea

that PPFs derive from nephric ridges is not new. It was already proposed by *Goodrich (1930)* based on purely anatomical observations. We suggest that the mesenchyme involved in this process arises from the PHMP coelomic epithelium and this process is dependent on WT1 and RA. Thus, WT1 has an evolutionary significance as a key factor involved in the origin of the diaphragm. The involvement of WT1 in the morphogenesis of the pleuropericardial septa had been described by *Norden et al. (2010, 2012)*.

In summary, Bochdalek's hernia could be regarded as an atavistic condition related with the failure in the ontogenetic/evolutionary process which replaced the thoracic intermediate mesoderm by lateral plate mesoderm expressing GATA4 under control of the G2 enhancer. WT1 was probably crucial in the process of generating the cellular substrate which made this replacement possible. A right/left asymmetry in this process is probably behind the left prevalence of this disease.

## Materials and methods

### Mouse lines and embryo extraction

The animals used in our research program were handled in compliance with the institutional and European Union guidelines for animal care and welfare. The procedures used in this study were approved by the Committee on the Ethics of Animal Experiments of the University of Malaga (procedure code 2015–0028). All embryos were staged from the time point of vaginal plug observation, which was designated as E0.5. Embryos were excised and washed in PBS before further processing.

The reporter Wt470LacZ mouse line is described in *Moore et al. (1998)*. The G2-*Gata4*$^{LacZ}$ and G2-*Gata4*$^{Cre}$ transgenic lines have been previously described (*Rojas et al., 2005*; *Delgado et al., 2014*). The Tg(Wt1-cre)#Jbeb (*Wt1*$^{Cre}$; MGI:5308608) mouse has been used in previous studies to trace or delete specific genes in WT1-expressing cells (*del Monte et al., 2011*; *Wessels et al., 2012*; *Cano et al., 2013*; *Carmona et al., 2013*; *Casanova et al., 2013*; *Cano et al., 2016*). For lineage tracing studies, heterozygote G2-*Gata4*$^{Cre+/-}$ mice and homozygote *Wt1*$^{Cre+/+}$ were crossed with *Rosa26R* (B6.129S4-Gt(ROSA)26Sortm1Sor/J, RRID:IMSR_JAX:003474) and *Rosa26*$^{EYFP}$ (B6.129X1-Gt(ROSA)26Sortm1(EYFP) Cos/J, RRID:IMSR_JAX:006148) mice to generate permanent reporter expression in G2-*Gata4* and *Wt1*-expressing cells. Generation of loxP-flanked *Wt1* mice were performed as described (*Martinez- Estrada et al., 2010*). Systemic *Wt1* knockout mice were generated by gene targeting, as previously described (*Kreidberg et al., 1993*).

### Immunohistochemistry, reporter analysis and in situ hybridization

For fluorescent reporter expression analysis and immunohistochemistry the embryos were fixed in 2% fresh paraformaldehyde solution in PBS for 2–8 hr, washed in PBS, cryoprotected in sucrose solutions, embedded in OCT (Tissue-Tek), and frozen in liquid N2-cooled isopentane. Samples were sectioned on a cryostat (10 μm) and cryosections were stored at −20°C until use. Other samples for routine histology were dehydrated and paraffin-embedded.

For in situ hybridization the embryos were isolated in ice-cold PBS and were fixed overnight in 4% paraformaldehyde in PBS. In situ hybridization was performed as previously described (*Wessels et al., 2012*). The Pax2 probe is described in *Dressler et al. (1990)*.

The monoclonal mouse anti-smooth muscle cell α-actin antibody (clone 1A4, Sigma, RRID:AB_476701) was used at a 1:100 dilution for immunofluorescence. Polyclonal rabbit anti-RALDH2 (a gift of Dr. Peter McCaffery, Eunice Shriver Center, University of Massachusetts) was used at a 1:5000 dilution. Polyclonal goat anti-GATA4 (Sc-1237, Santa Cruz, RRID:AB_2108747) was used at 1:50 dilution for immunofluorescence. Polyclonal chicken anti-GFP (13970, Abcam, RRID:AB_300798) was used at 1:200 dilution. Monoclonal anti E-cadherin (610181, BD Biosciences, RRID:AB_397580) and monoclonal anti-mouse WT1 (MAB4234, Millipore, RRID:AB_570945) were used at 1:100 dilution. For WT1, E-cadherin and GATA4 immunostaining, citrate buffer antigen retrieval was required. Negative controls were performed by incubation with non-immune mouse serum or rabbit IgG instead of the primary antibody.

### Image analysis

Image analysis was performed on digital images obtained from four *Wt1*$^{Cre}$;*R26R*$^{YFP}$embryos (two E11.5 and two E12.5). Between 6 and 9 Images of consecutive 10 μm thick serial sections covering

all the PHMP were processed by splitting the channels, transforming the green (YFP+) channel into a 8-bit format, and measuring the total area occupied by YFP+ cells in the left and right PHMP using Fiji image analysis software (RRID:SCR_002285). The total sum of the areas along all the serial sections was used as an estimation of the total volume occupied the YFP+ cells. We then calculated the R/L volume ratio for each embryo and this value was used for statistical comparisons.

## Rescue of phenotype with dietary administration of RA

Rescue of the phenotype in G2-$Gata4^{Cre}$;$Wt1^{fl/fl}$ embryos was performed by supplying RA in the diet of the pregnant female mice between E8.5 and E10.5, i.e. when the ST and PPFs develop. At 8.5 and 9.5 days of pregnancy, female mice were fed with a paste composed of 5 g of normal chow, 5 mL of distilled water, 1 mL of corn oil and 1 mg of RA (Sigma). Control females received the same diet without RA. The food was placed in a cage protected from light and pregnant female mice consumed the RA-supplemented diet normally. By the beginning of the day 10.5, RA-supplemented diet was replaced by normal chow. We collected the embryos at stages E13.5 and E15.5.

## Statistics

We used the Z-test for comparison of the R/L PHMP volume ratio and for comparison of the percentages of CDH incidence in RA treated and control embryos. For comparison of the size of the left pleural cavity opening in RA treated and control embryos we used the Student's test. In both cases, calculations were performed using Excel 2013 software.

## Acknowledgements

We thank Dr. John Burch and Dr. Brian Black for providing $Wt1^{Cre}$ and the G2-$Gata4^{LacZ}$ mice, respectively, Dr. Nick Hastie and Ofelia Martinez-Estrada for providing the WT1-LoxP mice and Wt470LacZ embryos. We also thank David Navas and Lina Molina for technical support and Dr. Peter McCaffery for the gift of the anti- RALDH2 antibody. This study was funded by grants BFU2014-52299-P (Spanish Ministry of Economy), ISCIII-RD12/0019-0022 (ISCIII-TERCEL), and P11-CTS-07564 (Junta de Andalucía). AR is the recipient of the grant PI14-00804 funded by Instituto de Salud Carlos III and cofounded by FEDER funding,

## Additional information

### Funding

| Funder | Grant reference number | Author |
| --- | --- | --- |
| Ministerio de Economía y Competitividad | BFU2014-52299-P | Rita Carmona<br>Ana Cañete<br>Laura Ariza<br>Ramon Muñoz-Chápuli |
| Instituto de Salud Carlos III | ISCIII-RD12/0019-0022 (ISCIII-TERCEL) | Rita Carmona<br>Ana Cañete<br>Elena Cano<br>Laura Ariza<br>Anabel Rojas<br>Ramon Muñoz-Chápuli |
| Consejería de Economía, Innovación, Ciencia y Empleo, Junta de Andalucía | P11-CTS-07564 | Rita Carmona<br>Ana Cañete<br>Elena Cano<br>Laura Ariza<br>Ramon Muñoz-Chápuli |
| Instituto de Salud Carlos III | PI14-00804 | Anabel Rojas |

The funders had no role in study design, data collection and interpretation, or the decision to submit the work for publication.

## Author contributions
RC, AC, EC, Acquisition of data, Analysis and interpretation of data, Drafting or revising the article; LA, Acquisition of data, Drafting or revising the article; AR, Analysis and interpretation of data, Drafting or revising the article, Contributed unpublished essential data or reagents; RM-C, Conception and design, Analysis and interpretation of data, Drafting or revising the article

## Author ORCIDs
Rita Carmona, http://orcid.org/0000-0002-9686-473X
Ramon Muñoz-Chápuli, http://orcid.org/0000-0001-6392-6802

## Ethics
Animal experimentation: The animals used in our research program were handled in compliance with the institutional and European Union guidelines for animal care and welfare. The procedures used in this study were approved by the Committee on the Ethics of Animal Experiments of the University of Malaga (procedure code 2015-0028). Every effort was made to minimize suffering of the mice used in our research.

## Additional files

### Supplementary files
• Supplementary file 1. Image analysis data of the volume taken up by the YFP+ cells in the right and left PHMP. This experiment is described at the end of the first section of the results.

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
