## [Decision Letter]

Thank you for submitting your article "Conditional deletion of WT1 in the septum transversum mesenchyme causes congenital diaphragmatic hernia in mice" for consideration by *eLife*. Your article has been reviewed by two peer reviewers, and the evaluation has been overseen by a Reviewing Editor and Harry Dietz as the Senior Editor. The reviewers have opted to remain anonymous.

The reviewers have discussed the reviews with one another and the Reviewing Editor has drafted this decision to help you prepare a revised submission.

Summary:

This is a thought-provoking study that deals with a topic of much clinical and fundamental interest. Congenital Diaphragmatic Hernia (CDH) is a common birth defect, affecting 1 in 3000 children. The most prevalent form, known as Bochalek-type CDH is characterised by a defect in the postero(dorsal) lateral area of the diaphragm which is asymmetric, the majority of cases affecting the left side.

There has been much speculation about the developmental origins of CDH, a problem that had been addressed in several recent papers using mouse models. Carmona and colleagues have investigated the role of the Wilms tumor transcription factor in septum transversum mesenchyme and identified a role in diaphragm development, with implications for both ontogeny and evolution of the diaphragm. They now provide the first animal model that appears to mimic the most common form of CDH. The authors trace the origins of the CDH to a deficiency of the posthepatic mesenchyme plate (PHMP) that arises from the coelomic epithelium. Importantly the authors show for the first time that the right part of the PHMP has more mesenchyme cells than the left at the time of septa development and this may explain the asymmetry. It has been shown previously by the authors that WT1 transcriptionally activates Raldh2 in the developing liver mesothelial derived cells. Raldh2 is essential for synthesis of retinoic acid (RA). In another study it was shown that maternally introduced RA can partially rescue CDH in a chemically induced model. Interestingly this chemical model also mimicked Bochdalek-type CDH, showing left side predominance. Carmona et al. show here that Raldh2 levels are much reduced in the mutant PHMP mesenchyme.

Overall, this is a scholarly descriptive study that had many interesting features. While the complex embryology of diaphragm development is carefully documented using a range of molecular and genetic markers, a number of major points need to be addressed to justify the authors' conclusions.

Essential points:

1) It would help to provide some marker analysis to inform on cell state (e.g. markers for proliferation/apoptosis, EMT states etc.). In particular, detailed analysis of EMT markers seems necessary to support the authors' arguments.

2) When do the embryos die and what other defects are observed? This is referred to very loosely.

3) The RA rescue is potentially important but very poorly described at the moment. First, rigorous statistical analysis has to be applied. Second, if not all animals are rescued could it be that some have partial rescue that indicates a qualitative difference between RA exposed and non RA exposed embryos? Has detailed anatomical analysis been carried out on all the embryos? It would help to provide some way of representing these data, perhaps a table with a few pictures of embryos.

4) Carmona et al. show here that Raldh2 levels are much reduced in the mutant PHMP mesenchyme. Supplementation of the pregnant females' diets with RA at 8.5 and 9.5 dpc seemed to result in rescue of the phenotype as 65% of animals with the mutant genotype have complete septation of the pleural cavity by E15.5. This would be a very important observation, but needs to be better documented.

5) Experimental evidence is required in support of cell migration from the coelomic epithelium to the PHMP or the PHMP to PFF. In the absence of direct evidence from DiI labelling, for example, comparison of Gata4 G2 Cre and conditional YFP expression may be helpful.

6) Is it possible to quantify the asymmetry in PHMP mesenchyme? Can the authors provide genetic evidence that this is connected to embryonic left right asymmetry?

7) How do the authors explain the variability in the conditional mutant phenotype? Do they consider that this is due to variable Cre expression or another cause? For example, variability in RA synthesis? Might removing one allele of Raldh2 influence the phenotype?

8) Please provide information as to the relative timing of the onset of Gata4 and WT1 expression in septum transversum mesenchyme.

9) Can the authors provide any data on WT1 expression in different species to support the evolutionary arguments in the eleventh paragraph of the Discussion?

---

## [Author Response]

*1) It would help to provide some marker analysis to inform on cell state (e.g. markers for proliferation/apoptosis, EMT states etc.). In particular, detailed analysis of EMT markers seems necessary to support the authors' arguments.*

We had performed a preliminary study using proliferation markers (PCNA and BrdU). We did not find clear differences, although a systematic study was not attempted. No signs of apoptosis such as pyknotic or fragmented nuclei were observed. The claim of a detailed analysis of EMT markers is very reasonable. Thus, we performed an analysis of E-cadherin expression in the left PHMP of four mutant and four control E11.5 embryos. E-cadherin is a good surrogate marker of EMT, since different factors (Snail, Twist, Zeb) can trigger EMT, but E-cadherin is always a target of them. The results are shown in the new Figure 5, and we think they clearly supports a downregulation of the EMT in the mutants.

*2) When do the embryos die and what other defects are observed? This is referred to very loosely.*

We have included a new table (Table 1) detailing all the genotypes and phenotypes. It is possible that some mortality occurs around E15.5, but the number of late embryos obtained is too little to confirm this. Other defects affecting cardiac development are described in Cano et al., 2016. Note that the number of mutant embryos studied has increased after the new experiments from 16 to 23, and the prevalence of diaphragmatic defects from 75% to 82,6%.

*3) The RA rescue is potentially important but very poorly described at the moment. First, rigorous statistical analysis has to be applied. Second, if not all animals are rescued could it be that some have partial rescue that indicates a qualitative difference between RA exposed and non RA exposed embryos? Has detailed anatomical analysis been carried out on all the embryos? It would help to provide some way of representing these data, perhaps a table with a few pictures of embryos.*

*4) Carmona et al. show here that Raldh2 levels are much reduced in the mutant PHMP mesenchyme. Supplementation of the pregnant females' diets with RA at 8.5 and 9.5 dpc seemed to result in rescue of the phenotype as 65% of animals with the mutant genotype have complete septation of the pleural cavity by E15.5. This would be a very important observation, but needs to be better documented.*

Yes, both criticisms are right. Thus, we have performed new experiments in order to study the development of the pleuroperitoneal septa at the stage E13.5, comparing embryos from females fed with the RA-supplemented diet and embryos from females fed with the same diet, without RA. The results are shown in the new Figure 7. We have shown a significant difference in the size of the left pleuroperitoneal discontinuity found in RA-treated and non RA-treated mutant embryos. This difference can be related with the lower prevalence of CDH in older embryos treated with RA. On the other hand, we have shown that the 5/8 proportion of normal diaphragm among the E15.5 embryos treated with RA in our first experiment is significantly different from the 4/23 proportion of normal diaphragms among all the non- treated mutant embryos (Z test=2,09, two tailed p-value= 0.037). All these data were obtained from anatomical analysis performed on serial sections of the embryos, not through direct observation of the phenotypes.

*5) Experimental evidence is required in support of cell migration from the coelomic epithelium to the PHMP or the PHMP to PFF. In the absence of direct evidence from DiI labelling, for example, comparison of Gata4 G2 Cre and conditional YFP expression may be helpful.*

Yes, direct labelling is really difficult to perform. However, we think that 1) the progression of cells from the G2+ domain towards the PPF is clearly supported by our description of the YFP+ cells that become more abundant in the PPF as the pleural cavities become closed and 2) this migration of cells had been described in human embryos by Hayashi et al. (2011). Since proliferation seems not to be a main role in the growth of the pleuroperitoneal septa, migration is the most probable explanation of this growth. Anyway, we have moderated the statements about the migration of cells, for example in the new version of the abstract.

*6) Is it possible to quantify the asymmetry in PHMP mesenchyme? Can the authors provide genetic evidence that this is connected to embryonic left right asymmetry?*

Yes, we did a quantification of the asymmetry by estimating the volume occupied by the YFP+ cells in the PHMP and calculating the R/L volume ratio. The right PHMP was 45% larger than the left one. The t-test for one sample compared with the expected value of 1 (symmetrical distribution) gave a result of 5,21 (p-value=0,0069 for three d.o.f.). Incidentally, we moved this paragraph from the mutant phenotype description to the first section of the results, where we describe the G2_Cre_;R26R_YFP_ embryos.

We think that there is a close relationship between the PHMP asymmetry and the main mechanisms regulating L/R embryonic asymmetry. We have recently collaborated in a manuscript about this issue that has just been submitted to Nature. It could be quoted in this context if accepted, but we cannot forward information in our manuscript at this moment.

*7) How do the authors explain the variability in the conditional mutant phenotype? Do they consider that this is due to variable Cre expression or another cause? For example, variability in RA synthesis? Might removing one allele of Raldh2 influence the phenotype?*

This is relatively frequent when using the Cre;LoxP technology. Sometimes the levels of Cre recombinase are not enough to produce full recombination. Alternatively, it is possible that the genesis of the CDH depends on a critical amount of cells in the PHMP, and small variations in this number of cells can lead to different outcomes. We also think that the levels of RA are critical, and this would explain the partial rescue of the phenotype just increasing the exposure of the embryos to this morphogen. So, yes, it is possible that Raldh2 hemizygous mice would be even more sensible to the conditional deletion of Wt1.

*8) Please provide information as to the relative timing of the onset of Gata4 and WT1 expression in septum transversum mesenchyme.*

Done. We have indicated in the text that Wt1 starts its expression in the ST/proepicardium by E9.0 and the enhancer G2 is active in this area about the same stage.

*9) Can the authors provide any data on WT1 expression in different species to support the evolutionary arguments in the eleventh paragraph of the Discussion?*

Not really. This is because our proposal of a colonization of nephric ridges by cells coming from the G2+ domain can only be applied to mammals, and we only have data on Wt1 expression and the activity of the G2 enhancer in mouse embryos. A way to support the evolutionary model would be to check if Gata4 expression in driven by a specific mesodermal enhancer in chick embryos, and what would be the anatomical domain where this enhancer is active.